Three-dimensional anatomy of the ostrich (Struthio camelus) knee joint

Chadwick Kyle P. kchadwick@rvc.ac.uk
Regnault Sophie
Allen Vivian
Hutchinson John R.
Structure & Motion Lab, Department of Comparative Biomedical Sciences, The Royal Veterinary College , Hatfield , United Kingdom
Wilson Laura
Electronic publication date: 2014 Dec 23
Publication date: 2014
Volume: 2
Electronic Location ID: e706
Received 2014 Oct 20; Accepted 2014 Nov 30
Copyright: © 2014 Chadwick et al.
Copyright year: 2014
Copyright holder: Chadwick et al.
License: This is an open access article distributed under the terms of the Creative Commons Attribution License, which permits unrestricted use, distribution, reproduction and adaptation in any medium and for any purpose provided that it is properly attributed. For attribution, the original author(s), title, publication source (PeerJ) and either DOI or URL of the article must be cited.
License URL: https://creativecommons.org/licenses/by/4.0/

Keywords: Morphology, Paleognathae, Ratite, Bird, Femorotibial joint, Patella, Sesamoid, Biomechanics

Funding: Leverhulme Trust RPG-2013-108 This work was completed with funding from the Leverhulme Trust (grant no. RPG-2013-108). The funders had no role in study design, data collection and analysis, decision to publish, or preparation of the manuscript.

==============================
The three-dimensional anatomy of the ostrich (Struthio camelus) knee (femorotibial, femorofibular, and femoropatellar) joint has scarcely been studied, and could elucidate certain mechanobiological properties of sesamoid bones. The adult ostrich is unique in that it has double patellae, while another similar ratite bird, the emu, has none. Understanding why these patellae form and what purpose they may serve is dually important for future studies on ratites as well as for understanding the mechanobiological characteristics of sesamoid bone development. For this purpose, we present a three-dimensional anatomical study of the ostrich knee joint, detailing osteology, ligaments and menisci, and myology. We have identified seven muscles which connect to the two patellae and compare our findings to past descriptions. These descriptions can be used to further study the biomechanical loading and implications of the double patella in the ostrich.

Introduction

Ostriches (Struthio camelus) are well known for their exceptional bipedal running abilities (e.g., Alexander et al., 1979). Their muscular, three-dimensionally mobile legs (Rubenson et al., 2007) are attractive subjects for studies of mechanical forces in the knee because they are able to accommodate large dynamic loads (Rubenson et al., 2011; Hutchinson et al., 2014). This large dynamic loading of the pelvic limbs in ostriches raise questions of how much support is active (i.e., muscular) vs. passive (skeletal, ligamentous, etc.) and the relative roles of the many structures in the knee region.

The knee joint sesamoid bones (kneecaps or patellae) in ostriches are of particular interest, because—unusually for birds and indeed all other animals—they are present as double (proximal and distal) rather than single bones. This has been recognized for at least a century and a half (Macalister, 1864; Haughton, 1864), however, until recently the double patellar sesamoids of ostriches have been overlooked in a comparative context. Regnault, Pitsillides & Hutchinson (2014) showed that ostriches are unusual among ratite (Palaeognathae) birds in having these two patellae—emus and other ratites (e.g., cassowaries, extinct moa) appear to have lost the patella completely, whereas kiwis and tinamous have retained a plesiomorphically small patella. The condition in other species is uncertain, and although Stannius (1850) and De Vriese (1909) hint of a double patella in Rhea, evidence so far indicates that the double patella evolved only once: within the lineage leading to Struthio.

All bones develop under a combination of genetic and epigenetic (mechanical) inputs, but sesamoid bones including the patella appear to be particularly dependent on early biomechanical influences (Sarin et al., 1999), with poor formation or complete developmental failure in embryonic immobilisation studies (Mikic et al., 2000; Osborne et al., 2002; Kim, Olson & Hall, 2009). Although the relative importance of biomechanical stimulation seems to vary amongst species and sesamoids (Vickaryous & Olson, 2007; Kim, Olson & Hall, 2009), it is clear that some aspect of normal embryonic movement is vital for sesamoid formation. Through the use of 3D models and finite element analysis, previous researchers have found that sesamoid ossification centres occur in regions of high tissue stress (Roddy et al., 2011). Moreover, it seems that the type of stress is key; high octahedral shear (i.e., pure shear) stresses appear to promote sesamoid ossification whereas hydrostatic (i.e., pure normal) stresses inhibit it (Sarin & Carter, 2000; Giori, Beaupre & Carter, 1993).

Multiple studies of real tissue in and ex vivo corroborate these ideas, and have found regions of fibrocartilage and bone proliferation corresponding with areas of compressional loading (Scapinelli & Little, 1970; Benjamin & Ralphs, 1997; Vogel & Peters, 2005). Potential mechanosensitive genes involved in the regulation of embryonic ossification have been identified (Nowlan, Prendergast & Murphy, 2008), though investigation of the role of genetics in sesamoid formation is complex and ongoing. Most researchers conclude that both mechanical and genetic factors interact and play complementary parts in creating sesamoid bones. In studying long bones, Carter, Mikić & Padian (1998) hypothesised that the relative rate of perichondral vs. endochondral ossification (controlled by genetics) in turn creates a biomechanical environment favourable (if the two occur simultaneously) or unfavourable (if perichondral precedes endochondral ossification) to the formation of secondary centres.

The aforementioned studies provide plausible explanations for the large variation of sesamoids across vertebrate taxa, even amongst those with grossly similar mechanical loading (e.g., birds; Regnault, Pitsillides & Hutchinson, 2014). However, more case studies are needed about the biomechanical function(s) that sesamoids perform in an individual at any one time, how those roles are moulded during ontogeny, and how the form and function of sesamoids evolve. In particular, it is not yet clear how much the sensitivity of sesamoids to their mechanical environment varies—are some, such as the patella, more phenotypically plastic in some taxa (e.g., birds) than others (e.g., mammals) (Barnett & Lewis, 1958)? If so, has any differential plasticity of sesamoid development played an important role in evolution, such as undergoing genetic assimilation (Sarin et al., 1999), or indicating key changes in locomotor function that correspond to altered loads on sesamoids (Hutchinson, 2002)?

We are interested in questions such as: within what tendons are the patellae developed and how are they loaded by the surrounding tissue? We aim to investigate these questions using a three-dimensional (3D) model of knee joint morphology. To do this, we scanned and digitally segmented the ostrich knee structures (muscles, ligaments, and bones) into discrete elements, allowing connecting tissues and muscles to be described and measured. We present an anatomical description of the morphology of the knee joint of ostriches, with a focus on features that are most relevant to the mechanics of the two patellae. Anatomy of the knee region has been characterized in numerous birds (Haines, 1942; Cracraft, 1971; Abourachid, 1991; Fuss & Gasser, 1992), and several studies contain descriptions of knee tissues in ostriches (Macalister, 1864; Haughton, 1864; De Vriese, 1909; Fowler, 1991; Bezuidenhout, 1999; Gangl et al., 2004; Wagner, 2004; Zinoviev, 2006; Smith et al., 2006). None of these studies focus directly on the two patellae and their interaction with surrounding tissues. The anatomy is not simple and is best understood with a full three-dimensional perspective; therefore we present an interactive 3D representation of a representative knee in a mature ostrich.

Materials and Methods

Our 3D model is based on the right leg of a skeletally mature adult (unknown age) male ostrich, body mass 71.3 kg. It was imaged with digital radiography (Fig. 1; Philips MX 8000 IDT 16 scanner, 16 bit images, ∼428 × 352 mm images at 10 pixels mm−1 resolution, varying kV and mAs), computed tomography (CT; Philips MX 8000 IDT 16 scanner, 120 kV, 100 mAs, 0.8 mm slice thickness), and magnetic resonance imaging (MRI; Philips Medical Systems Intera scanner, TR: 3596 ms, TE: 120 ms, α: 90°, 3 mm slice thickness). Following this the leg was dissected to qualitatively study the anatomy, and these observations validated through dissections of two other individuals (adults of unknown age, 65.3 and 130 kg body mass at death), CT imaging of six individuals (settings similar to those above), comparative studies of museum specimens (Table 1 and 15 others from various collections), and consultation of the relevant literature (cited above), to ensure that our observations on this individual were as representative as possible. All specimens were acquired from local farms after being euthanized for health reasons unrelated to this study.

Figure 1 Digital radiograph (A) and line drawing (B) of the ostrich right knee joint, in lateral view, showing sagittal plane locations of the two patellae with respect to bones and soft tissues.

Table 1 The measured lengths of the distal patella and femur in dissected and museum specimens, as well as the ratio of the two.

Subject	Distal patella length (mm)	Femur length (mm)	Ratio	
NHMUK (1888.3.15.1)	39.5	277.0	0.143	
NHMUK (1857.2.24.10)*	64.9	292.5	0.222	
NHMUK (1895.10.14.1)	57.7	306.0	0.189	
NHMUK (1925.5.12.1)	57.7	312.0	0.185	
NHMUK (1915.3.29.1)*	51.2	272.0	0.188	
NHMUK (1894.3.17.1)*	62.5	314.0	0.199	
NHMUK (1954.5.1)*	61.7	292.5	0.211	
NHMUK (1972.1.2)	44.9	284.0	0.158	
RVC1	33.1	306.0	0.108	
RVC2	75.7	317.0	0.239	
RVC3	67.2	307.2	0.219	
Average	56.0	298.2	0.187	
Standard deviation	8.4	14.8	0.024	
Notes.

Specimen RVC2 in italics is the main individual in this study.

* These specimens possess both femora and distal patellae; average lengths are shown. NHMUK specimen numbers pertain to osteological specimens held in The Natural History Museum, Tring, Hertfordshire, UK. RVC specimen numbers refer to specimens held at The Royal Veterinary College, Hatfield, Hertfordshire, UK.

To construct the model, we segmented the resulting DICOM images from CT and MRI for our representative ostrich specimen (and select comparative specimens) in Mimics (Materialise Inc., Leuven, Belgium) software. Bones were segmented from CT images and muscles were segmented from MRI images. The segmentations were individually and semi-automatically rendered into 3D objects. The 3D images of the bones were co-registered with the muscle MRI files and manually aligned to fit the limb posture from the MRI scan. This allowed 3D objects representing all relevant bones and muscles to fit together into a single model.

Results

The bones, muscles, ligaments, and menisci forming the 3D model created in this study can be viewed as a 3D PDF, hosted here http://dx.doi.org/10.6084/m9.figshare.1252187, or as individual STL files, hosted here http://dx.doi.org/10.6084/m9.figshare.1252166.

Osteology

The knee joint of an adult ostrich has five component bones: the distal femur, the proximal tibia and fibula, and the proximal and distal patella (Fig. 2B). The femur has asymmetric condyles, the lateral being appreciably larger than the medial. While we observed general asymmetry in other ratites, the differential size between lateral and medial condyle appears to be greatest in the ostrich. There is also a large lateral femoral epicondyle lateral to the lateral femoral condyle, forming a fibular trochlea. The tibiotarsus has lateral and cranial tibial crests extending from those two aspects of the proximal tibia.

Figure 2 Three-dimensional model of the ostrich right knee, showing bones, ligaments, and menisci.

LatCL (dark blue), lateral collateral ligament; MedCL (green), medial collateral ligament; CranCL (purple), cranial cruciate ligament; CaudCL (yellow), caudal cruciate ligament; meniscus (cyan). (A) Proximal view of ligaments, menisci, tibia, and fibula; (B) Cranial view of femur, tibia, fibula, proximal patella, distal patella, ligaments, and menisci.

The proximal patella lies high on the lateral femoral condyle, dipping slightly into the large sulcus in the position the specimen was scanned in (similar to that in three other specimens scanned). The proximal patella therefore topologically corresponds to the single patella of other birds, which occupies a position within or slightly above the sulcus (Shufeldt, 1884; Haines, 1942; Cracraft, 1971), and its flattened morphology likewise is similar.

The distal patella, which has only been briefly mentioned in literature (Macalister, 1864; De Vriese, 1909; Thompson, 1890; Bezuidenhout, 1999; Gangl et al., 2004; Wagner, 2004), is 75.7 mm long in our subject (ossified portion; Table 1) and sits in front of the lateral femoral condyle. There was considerable variability in the distal patella lengths we observed in our dissected and museum specimens, however, the length does roughly correlate with femur length (Table 1), which suggests a correlation with age. The distal patella extends down to just above the tip of the tibial crest, where it is connected by a short band of connective tissue (∼3 mm), apparently corresponding to a distal remnant of the patellar tendon. This distal-most sesamoid, much like the patella observed in some diving species (Shufeldt, 1883; Shufeldt, 1884; Thompson, 1890), appears like a proximal extension of the tibial crest, observed in other species such as Colymbus glacialis (Thompson, 1890; Vickaryous & Olson, 2007). Both patellae are enveloped by a thick, fibrous facial sheet to which many tendons contribute. While neither patella articulates directly with any other bone in the knee (through an articular cartilage interface), the layer of fibrous tissue between the femur and proximal patella is thin and may allow transmission of contact forces in some poses or loading regimes.

Ligaments and menisci

There are four primary ligaments which provide stability and alignment in the knee joint of ostriches, as in many other tetrapods (Fig. 2A; Fuss, 1989; Fuss, 1991; Fuss, 1996). A wide, flat collateral ligament spans the femorotibial joint space on either side, laterally and medially. The medial collateral ligament (MedCL) connects the medial femoral condyle to the tibiotarsus. It originates within a small fossa on the distal medial side of the medial femoral condyle and inserts distally to the tibial plateau on the medial edge of the proximal tibia. The lateral collateral ligament (LatCL) originates on the distal part of the lateral femoral epicondyle, and inserts onto the fibula, on the posterior-distal corner of the lateral side of the bulbous epiphysis, as well as the lateral meniscus, on the lateral side of the large, pointed cranial extension of the meniscus. There is additionally a larger lateral collateral ligament (LatCL_2) which originates at the very top of the lateral femoral condyle, inserts on the shaft of the fibula, and also connects cranially to the meniscus. The cranial cruciate ligament (CranCL) is round in cross-section, originates caudally (in the popliteal fossa) between the femoral condyles and inserts cranially on the tibial plateau (Fig. 3A). The caudal cruciate ligament (CaudCL) is thicker and flatter. It originates from a small impression on the medial side of the lateral femoral condyle, crosses over the top of the CranCL, and inserts on the caudomedial corner of the tibial plateau (Fig. 3A). We found slight differences in the size and shape of attachments, particularly on the tibial head, from previous descriptions (Fuss & Gasser, 1992); however, the general location agreed (Fig. 3B).

Figure 3 Cruciate ligament and meniscal insertion sites.

(A) Proximal view of the proximal right tibia and fibula, showing distal cruciate ligament and meniscal insertion sites (B) cruciate (speckled) and meniscal (solid) attachment sites on the distal femur (left and right columns) and proximal tibia (central column). Attachment sites shown in Fuss & Gasser (1992; top row) compared with what we observed (bottom row). Figure modelled after Fuss & Gasser (1992).

The medial meniscus sits between the medial femoral condyle and the tibial plateau. It is circular, thickest on its outermost aspect and thinner towards the incomplete centre, so that it forms a triangular wedge in cross section. Cranially, the medial meniscus connects to the lateral meniscus. The lateral meniscus is smaller than the medial meniscus and is longer (craniocaudally) than it is wide (mediolaterally). It sits primarily in the gap between the tibiotarsus and the fibula and extends cranially up the lateral femoral condyle. The femoral meniscal ligament attachments match well with Fuss & Gasser’s (1992) descriptions, however, the tibial attachments differ in size and two meniscal attachments on the fibula were found, which were not previously noted (Fig. 3B).

Myology

From our MRI scan and dissections of the knee region, we identified 12 distinct muscles that cross the knee joint near the patellae (see Hutchinson et al., 2014 for details on anatomical nomenclature): M. ambiens part 1 (AMB1), M. femorotibialis intermedius (FMTIM), M. femorotibialis lateralis pars distalis (FMTLD), M. femorotibialis lateralis pars proximalis (FMTLP), M. femorotibialis medialis (FMTM), M. fibularis longus (FL), M. gastrocnemius intermedius (GIM), M. gastrocnemius lateralis (GL), M. gastrocnemius medialis (GM), M. iliotibialis cranialis (IC), M. tibialis cranialis- femoral head (TCfem), and M. tibialis cranialis- tibial head (TCtib). Seven of these muscles are directly connected by tendons to the patellae: the IC, GL, FL, GM, FMTIM, FMTLD, and FMTLP (Fig. 4). The IC, FMTIM, FMTLD, and FMTLP muscle bellies lie proximal to the patellae, whereas the GL, FL and GM muscle bellies lie distally.

Figure 4 Representation of knee, in anterolateral view, showing superficial (A) and deep (B) muscles that attach to the tendofascial sheet containing the two patellae.

The knee extensors have major insertions around the patella. The IC originates on the cranial end of the ilium and inserts into the superficial tendofascial sheet as well as the medial side of the tibial head. The origin of the FMTLD is along the entire lateral femoral shaft and it inserts onto the deep tendofascial sheet above the lateral femoral condyle. The FMTLP origin occurs laterally on the trochanteric crest of the femur, and laterally on the proximal femoral shaft (Gangl et al., 2004), and inserts into the deep tendofascial sheet near the proximal patella. The FMTIM arises from the trochanteric crest and proximal three-quarters of the cranial femoral shaft (Gangl et al., 2004), and inserts on the deep tendofascial sheet, near the proximal patella and directly above the medial femoral condyle. A third femorotibial muscle, M. femorotibialis medialis, crosses the knee joint from the medial femoral shaft to the medial surface of the proximal tibia but does not interact with the patellar tissues and thus is not further described here (but see Zinoviev, 2006 for an accurate account and Hutchinson et al., 2014 for some explanation of confusion surrounding these femorotibial muscles’ identities in other literature).

The thin, round tendon of the AMB1, which originates from the pectineal (preacetabular) process of the pubis, runs through the tendofascial sheet and directly behind the distal patella in a distolateral direction, toward its fusion with the tendinous origin of M. flexor perforates digiti III, distal to the fibular head (Gangl et al., 2004). We observed that there were no direct tissue connections between the AMB1 tendon and the patellae. The tendon was free to slip and move independently within the tendofascial sheet, behind the distal patella, unlike in many other (neognath) birds in which the AMB1 tendon perforates or grooves the front of the patella (e.g., Shufeldt, 1884). However, we describe it here as it does run close to the patellae. The second head of M. ambiens, which is unique to ostriches, originates dorsal to the other from the cranioventral iliac rim and inserts on the medial surface of the proximal tibia (Hutchinson et al., 2014). However, it does not come near the patellae and so we do not further describe it.

The lower limb muscles of ostriches also have associations with the patella. The GL originates on the proximolateral side of the distal patella and superficial tendofascial sheet, joining with the GM and GIM into a single gastrocnemius end-tendon distally, and inserting onto the tarsometatarsus after wrapping around the intertarsal joint. The FL attaches to the distolateral side of the distal patella and splits into two tendons of insertion proximal to the intertarsal joint, Tendo lateralis and Tendo caudalis. Tendo lateralis inserts on the tendon of the M. flexor perforates digiti III, distal to the intertarsal joint, and the Tendo caudalis inserts on the lateral tibial condyle (Gangl et al., 2004). The GM takes its origin from the superficial tendofascial sheet and the medial side of the distal patella, and joins the gastrocnemius end-tendon before the intertarsal joint. Other lower limb muscles such as M. tibialis cranialis, M. popliteus and the many digital flexors do not originate near the patellae and are well described in the literature (Gangl et al., 2004; Zinoviev, 2006), so we do not discuss them here.

Discussion

We scanned, modelled, and dissected an ostrich knee and found previously undescribed or unclear morphology which may be crucial in ostrich knee function. Literature and other specimens bolstered our findings. From the detailed, three-dimensional anatomical data that we collected, we are able to confidently describe the functional attachments of muscles to the tendofascial sheets containing the two patellae, suggest mechanical implications of these attachments in a dynamic limb, and compare our findings to previous anatomical descriptions.

Functional attachments

To understand the loading across the knee joint of ostriches, identification and description of the tissues which directly interact with the two sesamoid bones is essential (Figs. 2–4). The two sesamoids are embedded in sheets of tendofascial tissue that bond together at the proximal edge of the distal patella. The sheets are the origins and insertions of various leg muscles. The proximal sesamoid constitutes the insertion of the FMTIM tendon proximal-medially and the FMTLP tendon proximal-laterally. Distally, the sheath attaches to the distal sesamoid. The distal sesamoid forms the insertion for the FMTLD tendon proximal-laterally and the IC tendon proximally. The distal sesamoid also is part of the muscle origins of the GL proximal-laterally, the FL distal-laterally, and the GM medially.

Based on our anatomical observations, we hypothesize that all seven of the muscles attached to the patellae would induce a compressive (into the joint) stress on the patellae. The GL, FL, and GM would also induce a tensile force component in the distal direction while the IC, FTIM, FMTLP, and FMTLD would exert a tensile force component in the proximal direction. This could create areas of large compressive and shear loading near the joint as the tissues wrap around the femoral condyles. In addition to the proximal-distal loading, additional mediolateral loads may occur as a result of the tissues wrapping around the distal femur where there is a complex surface between the large lateral condyle and deep sulcus. This variegated surface may induce higher stress concentrations in areas where the surface geometry is most irregular.

Patellar anatomy, connections and evolution

The first two (in 1894) mentions of ostriches having a second patella described it not as a true patella, but as an “ossified ligamentum patellae” (Macalister, 1864; Haughton, 1864). Since then there have been several other descriptions. Thompson (1890) mentioned a small double patella in ostrich while De Vriese (1909) described the patellae as large, consisting of two successive parts which connected to the tibia by a short ligament. Fowler (1991) claimed that ostriches only had a single ossified bone in the tendon which inserted onto the cnemial crest. We found this to be untrue of our adult subjects. In a juvenile ostrich hindlimb we observed joints which were not fully ossified and no patellae were detected, but it is still unclear when in ontogeny each patella ossifies.

In more modern literature, there have been two primary papers providing thorough descriptions of the patellae and related ostrich knee anatomy (Bezuidenhout, 1999; Gangl et al., 2004). Bezuidenhout (1999) describes a primary patella with medial and lateral articular surfaces, and a “second patella” distal to the patella. The “second patella” is described as a long bony column extending from the medial bottom edge of the patella to the tibial crest. This account was also the first to detail muscle attachments to the two patellae. Muscles described in association with the patellae were the M. femorotibialis medius (inserts laterally on the patella), the M. femorotibialis accessorius (inserts proximally on the patella), the M. femorotibialis internus (inserts medially on the patella), and the M. gastrocnemius (one of four heads originates from the patella). We consider the three M. femorotibialis to be what we describe as FMTLP and FMTIM (two muscle bellies, vide Zinoviev, 2006; Hutchinson et al., 2014). Gangl et al. (2004) first described what we consider another very important aspect of the double patellae, which is the tendofascial sheet in which they are embedded, connecting multiple muscles and inserting on the Crista cnemialis cranialis. The authors also detailed additional muscles which surround and attach to this tendofascial sheet. The muscles which Gangl et al. (2004) described as inserting onto the tendofascial sheet are the M. iliotibialis cranialis and the M. femorotibialis accessorius—caput mediale (what we describe as FMTIM). The tendon of the M. ambiens (here AMB1) was described as running medially to the proximal patella and caudally to the distial patella through the tendofascial sheet of the knee within its own channel. The M. fibularis longus and the M. gastrocnemius pars medialis were both described as having their origins on the distal patella.

Additional modern studies have also shed light on the patellae through both osteological (Wagner, 2004) and myological (Zinoviev, 2006; Smith et al., 2006) descriptions. Wagner (2004) added a description of the occurrence and shape of the proximal and distal patella of ostriches at various ages, as well as describing the fascia formed by the ends of multiple tendons which the two patellae are embedded. Zinoviev (2006) described the proximal patella as being embedded within the distal extension of the tendon of the M. femorotibialis medius pars distalis (what we describe as FMTIM), whereas Smith et al. (2006) described the patella to be within the flat tendon of insertion of the M. femorotibialis medius, externus, and accessorius (what we describe as FMTLP, FMTLD, and FMTIM, respectively).

The tibial crest (specifically, the cranial crest or crista cnemialis cranialis; see Regnault, Pitsillides & Hutchinson, 2014 for more discussion) in birds has been termed a traction epiphysis, or a projection from the end of a bone shaft with a separate centre of ossification to which a tendon inserts (Barnett & Lewis, 1958; Parsons, 1904; Vickaryous & Olson, 2007). Traction epiphyses derive, in an evolutionary sense, from sesamoid bones that have fused to the main bone of tendon/ligament insertion. The concurrent existence of a distinct patella and tibial crest does not falsify the hypothesis that the tibial crest is a traction epiphysis in birds, because the patella could be independent from whatever ancestral sesamoid fused with the tibia to form a traction epiphysis (Hutchinson, 2002). However, to our knowledge the ostrich is the only case among birds of a tibial crest and two patellae appearing concurrently (Barnett & Lewis, 1958). The most plausible hypothesis is that the distal patella in ostriches is a neomorphic ossification unique to that lineage, rather than an atavistic or plesiomorphic case of a lack of fusion of the traction epiphysis, because the evolutionary sequence of traits in birds is (1) appearance of a cranial cnemial crest in stem (ornithothoracine) birds (Hutchinson, 2002), (2) evolution of a patellar sesamoid in ornithurine birds (Regnault, Pitsillides & Hutchinson, 2014), and (3) origin of a second (distal) patella in ostriches (this study).

However, some birds have a proximally elongate tibial crest that has been proposed to be a patella fused to the proximal tibia (Shufeldt, 1883; Shufeldt, 1884; Thompson, 1890). The extreme proximity of the distal patella to the tibial crest in ostriches, while autapomorphic, presents an example of an intermediate condition between the ancestral lack of ossification in the distal patellar tendon and the derived state of a proximally extended tibial crest. Such evolutionary trajectories, however, have barely been studied in the lineages in which they may have occurred. Our study shows the need for careful re-examinations, using modern techniques such as 3D imaging, of phylogenetic patterns in the knee joint morphology of birds.

Conclusion

We have identified and described the tissues surrounding the knee joint in the ostrich and compared our findings to the previous literature. We have also speculated on the mechanics and functions of the anatomical features which directly interact with the patellae. It is still not clear why the double patellae develop in the ostrich, and if particular mechanical factors play a primary role in determining their shape and location. In future work, we intend to address these questions through modelling methods such as finite element analysis.

We thank Richard Lam and Renate Weller for their assistance with all of the scans. We also thank Jeff Rankin and Luis Lamas for sharing their expertise in ratite anatomy. Additionally, Joanne Cooper, Judith White and Hein Van Grouw at the NHM were very helpful in assisting our accessing of specimens, and Charlotte Brassey for providing access to CT scans of a southern cassowary and rhea. We thank the editor Laura Wilson and one anonymous reviewer in addition to reviewer James Neenan for their helpful critiques of the original manuscript.

Additional Information and Declarations

Competing Interests

Author Contributions

Animal Ethics

Data Deposition

John R. Hutchinson is an Academic Editor for PeerJ.

Kyle P. Chadwick conceived and designed the experiments, performed the experiments, analyzed the data, wrote the paper, prepared figures and/or tables, reviewed drafts of the paper.

Sophie Regnault contributed reagents/materials/analysis tools, wrote the paper, reviewed drafts of the paper.

Vivian Allen wrote the paper, reviewed drafts of the paper.

John R. Hutchinson conceived and designed the experiments, contributed reagents/materials/analysis tools, wrote the paper, reviewed drafts of the paper.

The following information was supplied relating to ethical approvals (i.e., approving body and any reference numbers):

All study material was cadaveric, therefore no ethical approval was needed.

The following information was supplied regarding the deposition of related data:

Figshare: http://dx.doi.org/10.6084/m9.figshare.1252166.

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
