# Peer review of "Three-dimensional anatomy of the ostrich (Struthio camelus) knee joint"

_PeerJ, doi:10.7717/peerj.706_

## Round 0.1 · original submission · Minor Revisions

Both reviewers suggest that this paper is suitable for publication following minor revisions. The authors have produced a detailed, well-structured and nicely illustrated paper on the 3D anatomy of the knee joint in ostriches. I have only minor editorial comments (see below), and I ask the authors to please pay careful attention to the helpful suggestions of both reviewers. Particularly, Reviewer#1 has requested some additional discussion on the identity of the elongated patellar crest (as mentioned by Barnett and Lewis 1958), and minor modification to the Introduction to make sure the aims refer only to work presented in this study (and not future FEA studies that may arise). Also, Reviewer #2 provides some important suggestions to improve the clarity of several figures, and I agree that these changes would be of benefit to the reader.

Ln14: “The pelvic limbs of ostriches raise questions..” this sentence is a little vague, what aspect specifically of the (study of the) pelvic limbs?

Ln26: I suggest removing “tantalizing”

Ln50: the link between phenotype and genotype (if tractable outside the realm of model systems) is extremely complex, potentially case-specific and much discussed from a theoretical perspective. To add this one sentence here is a gross over-simplification and I am not sure doing so adds anything given the context.

Ln60: what do you mean by ‘developmentally plastic’? How would you define and test that? The extent to which development is plastic (or conversely, constrained) in this context is unclear, and given the reference cited (that study has not experimentally manipulated conditions of lab-reared animals, for example), I suggest it may be better phrased here as phenotypic variation, or phenotypic plasticity.

Ln282: “various ages of ostrich”, better perhaps to say “…distal patella of ostriches at various ages”

Ln285: “while” should be “whereas”

Table 1, provide also the detail for RVC – Royal Veterinary College, UK – in the caption, as done for NHM,

Barnett & Lewis – the page range for the article is missing in the reference list – 593-601

Reviewer 1 ·

Basic reporting

A nice well illustrated paper about an interesting subject.

Experimental design

The question is well thought out and interesting. The design of the work uses the most modern techniques. I am a little worried that the question "how are the patellae of an ostrich loaded and how do they develop in response to these loads? " is not fully addressed. To truly answer the second part of this question use of FEA or a similar engineering approach could (or should) have been made. Still with a slight modification of the aims the paper is still a useful bit of work

Validity of the findings

I consider the data support the findings stated.

there are couple of dumb statements however:

This is unusual compared to humans (Rehder, 1983)

Who cares! The only useful evolutionary question is how does this compare to other ratites! - I consider this question must be answered in detail otherwise the research question is not properly addressed. The authors clearly have done some research on this: https://peerj.com/preprints/404.pdf

Another perhaps I note that the myology uses Baumel terminology I consider the osteology should to thus the distal patella should be patella dorsalis (Check Livezey and Zusi they no doubt had a "latin" name)

Additional comments

I would really like to see some discussion of Barnett and Lewis (1958) contention that the elongated patellar crest of some birds represents fusion of the patella with the patellar crest of the tibiotarsus to which the patellar ligament is attached in most birds but not ratites

·

Basic reporting

No Comments

Experimental design

No Comments

Validity of the findings

No Comments

Additional comments

This paper is a short but comprehensive anatomical description of the ostrich knee joint, with particular focus on the unusual double patellae found in this species. It is very well-written, well-structured, concise and appropriately referenced. Although I am not an expert on knee joints or ostriches, the anatomical description appears to be precise and thoroughly conducted, and is presented through clear and easily interpreted figures. My recommendations for improvement are thus extremely minor:
1. The title of the paper is somewhat grammatically flawed. A better and more concise title would be "Three-dimensional anatomy of the ostrich knee joint (Struthio camelus)" or "Three-dimensional anatomy of the ostrich (Struthio camelus) knee joint" depending on the editor's wishes.
2. The figures would all be greatly enhanced by the addition of scale bars.
3. Figure 1 would be clearer if an accompanying line drawing of the knee joint anatomy was included next to the radiograph image. Also state in the caption the view (right lateral?) and that it is the right leg.
4. Please add to the caption of Figure 2 a key for the different coloured areas and explain in full what each one represents (e.g. LatCL (dark blue) = lateral collateral ligament). Also that it is the right leg. B does not appear to be cranial view to me, more like anterior, is this correct? It also looks like this figure is intended to be small, owing to the large size of the letters "A" and "B" - I would suggest making it full page width.
4. In Figure 4, please use "A" and "B" instead of "right" and "left" to conform to your previous figure style. Also please state which view this is in.
Overall this is an excellent piece of work, and I look forward to seeing it, as well as future biomechanical studies, published.

---

## Round 0.2 · accepted · Accept

Thank you for carefully addressing all the reviewer comments in your revised version, and for placing materials used in this study in an online archive. The figures and discussion are much improved with this recent set of modifications, and I don't see any further points requiring attention. I'm happy to accept your revision and I look forward to seeing your paper published!